# Machine Learning-Based Plant Detection Algorithms to Automate Counting Tasks Using 3D Canopy Scans

**DOI:** 10.3390/s21238022

**Published:** 2021-12-01

**Authors:** Serkan Kartal, Sunita Choudhary, Jan Masner, Jana Kholová, Michal Stočes, Priyanka Gattu, Stefan Schwartz, Ewaut Kissel

**Affiliations:** 1Department of Computer Engineering, Faculty of Engineering, Cukurova University, Adana 01330, Turkey; skartal@cu.edu.tr; 2System Analysis for Climate Smart Agriculture (SACSA), ISD, International Crops Research Institute for the Semi-Arid Tropics, Patancheru 5023204, Telangana, India; S.Choudhary@cgiar.org (S.C.); J.Kholova@cgiar.org (J.K.); 3Department of Information Technologies, Faculty of Economics and Management, Czech University of Life Sciences Prague, Kamýcká 129, 165 00 Prague, Czech Republic; stoces@pef.czu.cz; 4Department of Electrical Engineering, Indian Institute of Technology Hyderabad, Sangareddy 502285, Telangana, India; ee18resch11007@iith.ac.in; 5Phenospex B. V., Jan Campertstraat 11, 6416 SG Heerlen, The Netherlands; s.schwartz@phenospex.com (S.S.); e.kissel@phenospex.com (E.K.)

**Keywords:** 3D point clouds, plant detection, machine learning, computer vision, phenotyping

## Abstract

This study tested whether machine learning (ML) methods can effectively separate individual plants from complex 3D canopy laser scans as a prerequisite to analyzing particular plant features. For this, we scanned mung bean and chickpea crops with PlantEye (R) laser scanners. Firstly, we segmented the crop canopies from the background in 3D space using the Region Growing Segmentation algorithm. Then, Convolutional Neural Network (CNN) based ML algorithms were fine-tuned for plant counting. Application of the CNN-based (Convolutional Neural Network) processing architecture was possible only after we reduced the dimensionality of the data to 2D. This allowed for the identification of individual plants and their counting with an accuracy of 93.18% and 92.87% for mung bean and chickpea plants, respectively. These steps were connected to the phenotyping pipeline, which can now replace manual counting operations that are inefficient, costly, and error-prone. The use of CNN in this study was innovatively solved with dimensionality reduction, addition of height information as color, and consequent application of a 2D CNN-based approach. We found there to be a wide gap in the use of ML on 3D information. This gap will have to be addressed, especially for more complex plant feature extractions, which we intend to implement through further research.

## 1. Introduction

There is an increasing demand from plant-related research disciplines to access the specific plant parameters in large numbers of plants with enhanced throughput. This has become possible with the development of novel technologies, including sophisticated sensors and state-of-the-art computer vision algorithms (i.e., plant phenomics) [1,2,3,4,5]. These technologies provide powerful scanning tools (e.g., [6,7,8]) but usually lack the development of algorithms that can detect individual plants in complex canopies or calculate complex plant features (reviewed, e.g., [1,2,3]). Therefore, more flexible approaches are required to meet the growing demand for detailed plant observations, notably, automated pipelines that extract complex plant features (e.g., to develop climate-ready crops, among others) [1,4,9,10,11,12].

One example of such efforts is a phenomics platform LeasyScan (LS; Phenospex, Heerlen, The Netherlands; https://phenospex.com/; accessed: 20 August 2021) constructed at ICRISAT (http://gems.icrisat.org/leasyscan/; accessed on 21 August 2021; Vadez et al., 2015) with the primary purpose of facilitating research and development of climate-ready crops (e.g., [13,14,15,16,17]. For this purpose, the system is constructed outdoors and uses a set of mobilized sensors (“sensor-to-plant” principle) for continuous plant monitoring at scale (regular throughput is 4800 mini-plots per 2 h; [5]). This level of automation is technically feasible with laser scanning technologies such as PlantEye^®^. In brief, the PlantEye^®^ laser-based system projects a thin laser line in the near-infrared (NIR; 940 nm) region on plants and reconstructs the 3D-canopy surface based on the reflected light captured (the physical equipment setup is illustrated in Figure 1, also see Figure 2 in Vadez et al., 2015). PlantEye^®^ computes a set of basic canopy parameters within the predefined space through deterministic methods similar to those in Fanourakis et al. [18]. In principle, the algorithm triangulates the 3D point cloud into a mesh, out of which a subsequent algorithm computes leaf area, plant height, and a few indicators of leaf area distribution in the predefined space.

While capturing high-resolution 3D point clouds that could unlock the required traits (such as leaf area and individual plant height), plant segmentation becomes the real issue. An elegant way to break this analytical bottleneck might be to integrate machine learning (ML) approaches into existing data processing pipelines.

Pound et al. [19] predicted the paradigm shift in phenotyping aided by deep learning approaches. Since then, ML has demonstrated its potential in various automated agricultural systems ([20,21,22,23,24,25,26,27,28,29,30,31,32]). Furthermore, Kamilaris and Prenafeta-Boldú [33] reviewed studies that used ML algorithms in plant and agricultural research areas emphasizing that those algorithms provide high accuracy and outperform deterministic image processing techniques. This evidence encourages the exploration of ML approaches for the construction of an appropriate pipeline for PlantEye-generated data. While ML offers the potential to provide generic solutions to plant image analysis problems, it also brings some challenges. For example, although ML algorithms achieve state-of-the-art performance in 2D images, these are generally not used for 3D point-cloud data, typically produced by laser-based technologies.

Additionally, only a few studies have been conducted on 3D point cloud data to identify single plant traits, such as the single plant canopy, flowers, and panicles [34,35,36]. Itakura and Hosoi [34], focused on the point clouds in a specific height range (0.5 to 1.5 m) and detected each cluster of tree trunks based on point density. Similarly, Malambo et al. [35], detected individual sorghum panicles using a density-based clustering algorithm, namely DBSCAN. In another study, Sun et al. [21] detected the cotton bolls. The most significant limitation is that all these studies were based on data point density and the morphological characteristics of the plants. Moreover, all of these studies utilized clustering (unsupervised learning) algorithms, and none of them utilized supervised learning algorithms, which can learn wider range of morphological characteristics of the plants from the data. Approaches that do not use supervised ML techniques (e.g., clustering algorithms) are inadequate to analyze plants that have different phenotypes. For example, mung bean can form very different architectures depending on the environmental conditions, and the same variety of mung bean can exhibit different numbers and sizes of plant organs (leaves, branches). Therefore, analytical plant data research needs algorithms that can learn all of the different morphological characteristics of a plant species and detect plants without relying on assumptions (i.e., number of leaves, size).

Compared to 2D machine algorithms, the number of ML algorithms developed to analyze 3D point cloud data is limited in the literature. The main reason for the research gap in this area is the lack of public benchmark datasets and the difficulty of labeling 3D objects, limiting the further application of ML algorithms on 3D point cloud datasets. Other factors widening this gap might include the unordered format of point clouds data and the associated processing costs. At this research stage, applying these emerging methods could also be complex for ‘real-life’ scenarios where sensors could create imperfect object reflections (e.g., due to overlapping).

Significant improvements in point cloud plant data analysis are needed to keep pace with developments in imaging and sensor technologies [1]. Collectively, these improvements will accelerate the selection of a new generation of crop cultivars that are more adapted to climate change. Although it is already feasible to gather data on hundreds of plants to facilitate the selection, data handling and processing is still a significant challenge when translating sensor information into knowledge. Considering these, a novel pipeline that could help researchers transform large numbers of images and sensor data into knowledge is proposed in this study.

The study intended to take advantage of the recent development in ML approaches and attempt to build alternative and more flexible PlantEye data processing architecture. The main objective was to automatically segment the individual plants from complex canopy scans and set the base for extracting features from individual plants in future applications. After the individual plant detection, its application in the plant counting tasks has been prioritized, as this is currently performed manually with high resource demand (e.g., time), which is also prone to human error.

Contrary to the 3D point cloud data analysis, the studies reviewed above, 2D super-vised ML algorithms, which can learn the different morphological characteristics of a plant species and detect plants without relying on assumptions, were employed in the proposed pipeline. The reason 3D ML algorithms were not utilized in this pipeline is discussed in detail within the discussion section.

Technically, we investigated whether and how ML-aided data analytics could utilize 3D point clouds with minimal data processing cost for automatic detection, counting the number of plants in an experimental space. This included (1) development of a 3D point cloud segmentation algorithm to separate plants from their surroundings; (2) development of ML-based mung bean and chickpea plant detection models; and (3) evaluation of the proposed pipeline performance.

## 2. Materials and Methods

### 2.1. Plant Material and Cultivation Conditions, Ground Truth Measurements

Two sets of experiments were undertaken at a high-throughput phenotyping platform–LeasyScan in 2019. A single plant genotype was planted in one sector consisting of a PVC tray of (length x width x height) 64 × 40 × 42.5 cm for mung bean and 54 × 36 × 34 cm for chickpea, containing 50–60 kg of *Vertisols* collected from the ICRISAT farm. As each sector (tray)contains a single genotype in regular experiments, it is necessary to separate the sectors in any data processing pipelines (see Section 2.3; Step 1). The first experiment (Exp. 1) included 52 elite mung bean cultivars (*Vigna radiata*) from different geographical locations in India collected from the WorldVeg Center, Hyderabad, India. It was planted on 20 December 2018 in 3 replications, and 0–4 plants per sector were maintained after 18 January 2019. The plant numbers in each sector were counted manually on 19 January 2019, considered as ground truth data. The 3D point clouds data were obtained on the same day. The second experiment included 2800 chickpea germplasm lines (*Cicer arietinum*) planted on 11 March 2019, and 0–8 plants per sector were maintained from 26 March 2019. Manual plant counting in each sector was recorded on 27 March 2019, and 3D point cloud reflections of plants for analysis were gathered from the same day.

### 2.2. Basic Setup and Scanning Protocol

The LeasyScan platform Planteye F300 (Phenospex B.V., Heerlen, the Netherlands) projects a very thin laser line in the near-infrared (NIR) region of the light spectrum (940 nm) on plants and captures the reflected light with an integrated CMOS camera. A camera with a 45-degree angle then captures the reflection at a high rate (50–80 pics·s^−1^), allowing the reconstitution of 3-D images. The scanners measure pre-set areas (so-called sectors, one sector was planted with a single genotype). A grid of metal barcodes throughout the platform allows identification of the start and stop of each sector and identification of its ID. Furthermore, the barcodes reference the scanner position in the y and z directions. The scanners are moved at a constant speed of 3 m·min^−1^ while scanning and 8m·min^−1^ when returning. A scanner supporting device (SSD) scans the entire platform, 4800 sectors, in two hours. During the scanning process, the scanner linearly moves over the plants and generates 50 height profiles per second, which are then automatically merged into a 3D point cloud with a resolution of around 0.8 × 0.8 × 0.2 mm in the XYZ direction coordinates, respectively. Each scanner sends its data wirelessly to a server where a range of plant parameters are computed. These parameters can be visualized through a web-based interface called HortControl. In addition, the platform is equipped with a set of 12 environmental sensors (Campbell Scientific, Logan, UT, USA) that continuously monitor relative humidity (RH%) and temperature (T °C), integrating values every 30 min, one light sensor, one wind sensor, and one rain gauge. The environmental conditions can also be visualized in HortControl. Details about the equipment can be found in Vadez et al. (2015). Currently, algorithms based on deterministic approaches are in place to process 3D point clouds for whole plant parameters (leaf area, plant height, and point-cloud distribution; [5]).

### 2.3. Tray Segmentation (Step 1)

The raw point clouds data collected outdoors by the LeasyScan platform contain trays and soil (see Figure 2). Separation of the trays (sector) from each other was a necessary preprocessing step for soil segmentation, as the soil could not be separated from the plants by a fixed height coordinate for the reasons outlined in Section 2.4. Therefore, firstly, the initial raw data file consists of 12 trays divided into 12 equal parts along the y axis (the axis the trays were lined up). Partial views of the sample scan files, and separated tray data are given in Figure 2 (mung bean).

Second, the frame data of the tray was deleted, and the soil and plant parts of the data were saved as a separate data file. Figure 3 illustrates the result of the tray-cleaning process for mung bean (left) and chickpea (right) files.

### 2.4. Soil Segmentation (Step 2)

The soil segmentation aims to find the ground plane in the point cloud and separate it from the plants. Although the current point cloud data have a height value, this could not be performed satisfactorily due to non-uniformity in soil levels. The amount of soil in the middle of the tray was relatively lower than the sides, and there was also a slight difference in soil level when compared between trays. Therefore, when soil and plants were separated with a fixed height value, too much soil data remained on the sides, or too much plant data were eliminated (see Figure 4). In particular, if there were plants that germinated late and were closer to the soil in a box, a large part of the data for these plants were lost in the fixed-point separation process.

More complex algorithms have been decided to use to solve this problem and obtain clear and robust data. Therefore, this process was carried out with Region Growing Segmentation (RGS) and Random Sample Consensus (RANSAC) algorithms, which are widely used in the literature [36]. The steps of the algorithms and the obtained results are explained in the following sections.

#### 2.4.1. Region Growing Segmentation (RGS)

The objective of the RGS is to merge points that are close enough to each other in terms of their smoothness value. The work of the algorithm is based on the comparison of angles between the normal points. Clusters of points that are considered to be part of the same smooth surface are produced as output from the algorithm. First, the normal of each point was calculated based on other points around it. Second, all points are sorted by their curvature value. Then, the algorithm accepted the points with small curvature values as initial seed points and started the region’s growth. The flowchart of the RGS algorithm [37] is shown in Figure 5.

The most significant advantage of the RGS method is that it can adequately separate the regions that have the same properties. The disadvantage is that it is sensitive to noise. After the RGS method was executed, the data cluster having the most data points was assumed as “soil data” and separated from the others. Outputs of the RGS algorithm are given in Figure 6 for a comparison to the following method.

#### 2.4.2. Random Sample Consensus (RANSAC) Model

The RANSAC model was used to estimate the parameters of a mathematical model from a data set. The algorithm assumes that the dataset we are working with contains both inliers and outliers. The data that fit the specified model are called inliers, while the data that do not fit are called outliers. RANSAC randomly samples a subset of points at each iteration. The selected subset is considered to be inliers, and this hypothesis was tested in every iteration. The flow chart of the RANSAC algorithm is given in Figure 7 [38].

An advantage of the RANSAC algorithm is that it can estimate a model with a high degree of accuracy even if a significant number of outlier points are contained in the data set. A disadvantage of RANSAC is that there is no upper bound for the time required to calculate the model. When the algorithm is terminated at the end of the specified iteration, the found model may not be a model that fits well with the data. Furthermore, RANSAC can only estimate one model for a given dataset at a time. In our study, soil surface data are considered inliers, and vegetation pixels are considered outliers. Outputs of the RANSAC algorithm are shown in Figure 6.

When the results produced by the soil segmentation algorithms were examined in detail, it was visible that both algorithms produced almost the same results and successfully separated the plant data from the soil data. As the soil area was not flat enough, some soil data appeared in the results of the RANSAC algorithm. Considering the advantages and disadvantages of the algorithms, the RGS algorithm was preferred for the soil separation process in the present study.

### 2.5. Rasterization (Step 3)

In this step, the pure plant data obtained from the previous step were converted into 2D images. The purpose of rasterization into 2D format was well described in the Introduction and Discussion sections. Before the rasterization process, a Statistical Outlier Removal (SOR) algorithm was applied to filter out the noisy data (outlier) [39] . Then, the projections of the data points on the two-dimensional x- and y-planes were calculated. The only color information value that the data set contains is “intensity”, which is the intensity of the laser reflection received by the camera. However, the laser does not have the same color quality as a regular light source. The reflection can be very diffuse, hence its biological use is always questionable. Therefore, each pixel was colored according to the height gradient of the relevant point-cloud data. When the rasterized images in Figure 8 were examined, it can be seen that relatively meaningful images have been formed using the height value.

### 2.6. Plant Detection and Counting (Steps 4–5)

#### 2.6.1. Convolutional Neural Networks (CNNs) for Object Detection

Traditional CNN models contain several to hundreds of layers of convolutions. After sufficient training, the model becomes suitable for recognizing features in images and classifying them. However, real-world images contain multiple objects and it is often desirable to detect these objects separately. Therefore, object detection models require Region Proposal Networks (RPNs) to predict object locations.

An RPN is a CNN that hypothesizes object bounds and objectness scores at each position. Faster RCNN [40] is one of the well-known algorithms in this field. This algorithm is built on the CNN architecture, and it additionally can detect multiple objects in an image. A key idea of this algorithm is to merge RPN and Faster RCNN (Girshick, 2015) into a single network, allowing them to share a common set of convolution layers, thereby reducing the overall complexity and training time. A general view of the Faster RCNN architecture used in this research was given in Figure 9. Detailed information about the architecture can be found in [40] and briefly described.

The Faster RCNN model consists of two modules, a proposer and a detector. The proposer module takes an image as input and produces a set of rectangular object proposals, each with an objectness score. Then, the detector module uses the proposed regions for object detection. At the first step of the Faster RCNN, an RGB image is processed through a pre-trained CNN (feature extractor) to obtain a convolutional feature map. The resulting feature map is used in both proposed and detector modules.

In this study, Faster RCNN architecture combined with two modern feature extractors (Inception-v2 and Resnet50) to construct two different plant detection algorithms. In-ception-v1 (GoogleNet), the winner of the ImageNet Large-Scale Image Recognition Competition 2014 (ILSVRC14), was proposed by Google in 2014. The layout of the Inception-v2 network (number of convolutions, pooling, and inception modules and their order) can be found in [41]. Similarly, ResNet architecture was the winner of the ILSVRC15, which was proposed by the Microsoft Research team in 2015. The 50-layer Resnet, namely ResNet50, was utilized as the feature extractor in this study.

Considering the Faster RCNN model’s working principles, the differences between the adjacent pixel values in any part of the image may cause random patterns to occur on the image, resulting in region proposals in the related area. Particularly, in plant detection systems, regular operations (e.g., sowing, weeding, mulching, and thinning) on cropland result in dynamic changes of the soil surface, increasing the possibility of random patterns occurring. In the ongoing process, if the existing pattern in the proposed area is similar to a plant, it may cause false detection. Additionally, the random patterns formed by the soil surface in the background of the plants make it difficult for the CNN model to learn the plant pattern correctly. For these reasons, in object detection algorithms, clearing the areas where the object does not exist, namely the backgrounds, is critically important in terms of minimizing false detections [42,43]. Taking this into account, a great emphasis has been placed on properly clearing background data from existing images in this study.

#### 2.6.2. Image Labeling and Dataset Production

The performance of CNN depends on the quality and size of the datasets more than other machine learning algorithms. The size of the training data set must be large enough to prevent an overfitting problem. Overfitting is the case where the model learns the training data too well and produces good results on the seen data (training set) but performs poorly on the unseen data (test set). Having a large dataset is one of the simple ways to prevent this problem. Furthermore, the data set must include all conditions, such as various mung bean growth stages and shadows caused by neighboring mung beans, to improve robustness [32] Accurate labeling is critical for efficient image search and retrieval.

The images in the data set were taken after a thinning process that corresponds to the 18th day after the mung beans and chickpeas started germinating. The 365 images of mung bean and 1141 images of chickpea with different states, sizes, and shapes were selected. Then, the manual labeling was conducted by experts. Bounding boxes were drawn, and the individual plants in each image were labeled. Positive samples with insufficient or unclear pixel areas were not labeled to avoid overfitting problems in the CNN. Examples of labeled images are shown in Figure 10. 

#### 2.6.3. Training and Models Configuration

Two special versions of the Faster RCNN model were selected to be used in the present study. These models are ‘Faster RCNN Inception-v2’ and ‘Faster RCNN Resnet-50’. Inception and Resnet are special CNN models employed as feature extractors in Faster RCNN models, respectively. The models employed were pre-trained on the COCO (Common Objects in Context) dataset. Thus, selected models were only fine-tuned with training images of mung bean and chickpea plants. Overall, 20% of the mung bean and chickpea images were used as test sets, whereas 80% were used as training and validation sets. As mung beans and chickpeas were grown in spatially and temporally separate experiments, the two separate models specialized in recognizing mung bean or chickpea species were trained to increase the accuracy. The models were trained for 10,000 epochs on Windows 10 with Intel Core I7 3.5 GHz processor and NVIDIA GeForce GTX 1070 (8 GB) GPU.

In order to further improve model quality and its performance, all parameters were carefully selected according to the characteristics of the used dataset and the selected models. The “aspect ratios” parameter defines ratios for sides of a proposed regions as 0.5, 1.0, and 2.0. The “iou threshold “parameter was set to 0.2 to make proper filtering for overlapping boxes. Similarly, the “score-threshold” value was set to 0.5 to eliminate those proposals that are most likely to be incorrect. In a default configuration, it is set to a value close to 0, meaning that all proposals are accepted. As we have a limited number of plants in each tray, the “max detections per class” parameter was set to 10.

### 2.7. Performance Evaluation

After training the CNN model as described in Section 2.6.3, the detection of plants was performed on the 2D images. As the previously trained model was used in this step, the required processing cost and time were relatively low.

The performance of our plant detection model was evaluated according to the proposed challenge detection metrics for the COCO dataset, namely mAP_COCO_, that are similar to Pascal VOC metrics but have a slightly different implementation, and it also reports additional statistics such as mAP (mean Average Precision) at IoU (Intersection over Union) thresholds of 0.5:0.95, and precision or recall statistics. A mAP curve with values ranging from 0 to 1 was used to evaluate plant detection performance. IoU is used to decide whether a prediction is correct. A prediction is considered false positive (FP) if IoU < threshold, and true positive (TP) if IoU > threshold. IoU is defined as the area of the intersection divided by the area of the union of a predicted bounding box (*B_p_*) and a ground-truth box (*B_gt_*):(1)$$IoU=\frac{area\left(B_{p}\;\cap \;B_{gt}\right)}{area\left(B_{p}\;\cup \;B_{gt}\right)}$$

While Precision (P) measures how many of the detected objects are, in fact, mung bean or chickpea plants, Recall (R) measures algorithm ability to locate mung bean and chickpea plants within images. Therefore:(2)$$P=\frac{TP}{TP+FP}=\frac{TP}{\#predictions}$$
(3)$$R=\frac{TP}{TP+FN}=\frac{TP}{\#ground-truths\;}$$
where FP: False-Positive and TN: True-Negative. FP values represent the existence of the plant not found in the ground-truth table. TN is the model’s inability to detect an existing plant.

However, for each IoU value, the counts of the TP, TN, FP, FN terms and the mAP values calculated accordingly change. One way to overcome this problem is to use a range of IoU threshold values (0.5:0.95), calculate mAP for each IoU, and take their average to find the mAP_COCO_ as in Equation (4).
(4)$$mAP_{COCO}=\frac{mAP_{0.50}+mAP_{0.55}+\ldots \;mAP_{0.95}}{10}$$

In this study, the *mAP_COCO_* metric is used to compare the performance of employed models. According to the purpose of the study, the accuracies of the proposed pipeline were evaluated by comparing the predicted plant counts of each method with those made manually in the plant growing field itself and calculating the Mean Absolute Percentage Error (MAPE):(5)$$MAPE\left(\%\right)=\frac{\sum_{i=1}^{n}\left(\left|\frac{C_{predicted}-C_{manual}}{C_{manual}}\right|*100\right)}{n}$$
where n is the total number of trays, C_predicted_ is the number of plants automatically counted, and C_manual_ is the real number of plants manually counted in the growing field. 

## 3. Results

This study provides an automated pipeline to detect and count mung bean and chickpea plants using the application of computer vision on 3D point cloud data. The whole pipeline is based on five main steps. While the first three steps consist of preprocessing operations to clean the data and make them suitable for computer vision algorithms, the fourth step utilizes the trained model (Inception-v2 or ResNet50) to detect the plants. The entire process is illustrated as a flow chart using mung bean images and is outlined in Figure 11.

In the first step of the “Plant Counting Pipeline”, the raw data file was read from the test data folder and divided into the number of trays at equal intervals along the y coordinate. Then, the frame data of each tray was cleared using the trays’ information such as width, length, and edge width. In Step 2, the non-plant area was removed by detecting the soil surface using the RANSAC segmentation algorithm explained in Section 2.4.2. In Step 3., the obtained pure 3D plant data were converted into 2D images on which traditional CNN algorithms can be applied. Height values were used for color gradient. The training process of the models was performed only once to train the network. Once the model was trained, it was used continuously for plant detection in Step 4. Therefore, the 2D images were fed into the trained CNN model, and plant detection was performed. Finally, the total number of plants for each tray was calculated.

The steps shown in Figure 11 work similarly, both for the chickpea and mung bean plants. The algorithm parameters were tuned according to the features of the dataset.

Two different pre-trained state-of-the-art object detection models were fine-tuned for mung bean and chickpea datasets over 10,000 epochs. The overall performance (*mAP_COCO_* scores) of the ‘Faster RCNN ResNet50′ and ‘Faster RCNN Inception-v2′ models for both the mung bean and chickpea data sets are shown in Figure 12.

Regarding *mAP_COCO_* (Figure 12), Inception and ResNet models have similar performance for detecting mung bean and chickpea plants. In addition to Figure 12, the reached *mAP_COCO_* values are shown in Table 1. In order to compare and interpret the study’s success, the *mAP_COCO_* values of these algorithms on the commonly used COCO dataset are also given. 

Comparing the *mAP_COCO_* scores obtained in this study with the scores on the COCO data set of these models shows the differences between the success scores. These results quantitatively reveal how successfully mung bean and chickpea detection models were trained within this study. The most significant factor for the models to achieve such a high success compared to the classical data set is that the existing data have been successfully separated and cleaned as described in Section 2.3 and Section 2.4. Thus, different object patterns on the pictures were prevented, and possible false detections were minimalized. 

To evaluate the performance of the plant counting pipeline under the aim of this project, the number of test images, the number of plants counted manually, and the number of plants predicted by models is given in Table 2. The best prediction results are pointed out in bold. According to the best prediction counts (287, 1620) and the MAPE results (6.82 and 7.13) given in Table 2, mung bean plants and chickpea plants were predicted with 93.18% and 92.87% accuracy, respectively. The results indicate that the models achieved good performance in terms of MAPE for both legume species.

The models were applied to eight randomly selected test images to visualize the results and examine them in detail. The results produced by each model and the original view of the four mung bean and four chickpea test images are given in Table 3.

When the test images were examined in detail, some noise and barcode data were seen. However, as the patterns of these small amounts of data are not similar to the plant patterns, it does not cause any significant error. Additionally, there were slight overlaps on plants, but the models still worked successfully. Specific errors were observed when plants were too large or too close, e.g., the ResNet model detected extra mung bean plants in Image 4, and the Inception-v2 model missed a chickpea in the final test image. Apart from mislabeling, it is seen that almost all of the mung beans and chickpeas in the test images were detected correctly. The results indicated that the models are robust against irrelevant data patterns (barcode or noise), although overlapping plants may cause a particular error. As a result, considering the performance metrics given in Table 1 and Table 2, the mung bean-Inception-v2 and chickpea-ResNet50 models were chosen for the application and future development to automate the plant counting activity.

Training time and plant counting time of the algorithms are given in Table 4. The training time indicates the time required for training the Faster RCNN models. ‘Faster RCNN Inception-v2′ and ‘Faster RCNN ResNet50′ models required 1.25 and 1.45 h for the training process, respectively. The training times required by the models are reasonable, considering the training can be delivered offline and does not have to be repeated often.

The plant counting time includes all the time required along the proposed pipeline (tray segmentation, soil segmentation, rasterization, plant detection on trained model, and counting). ‘Faster RCNN Inception-v2′ and ‘Faster RCNN ResNet50′ models required 15 and 18 s, respectively. This plant detection time is fast enough to analyze the data produced by LeasyScan. Accordingly, 240 raw mung bean data files, which corresponds to 2880 trays, can be processed within 1 h. On the other hand, it is possible to achieve better (shorter) training and testing times with better hardware than the existing one (Intel Core I7 3.5 GHz processor and NVIDIA GeForce GTX 1070–8GB).

## 4. Discussion

### 4.1. Overview

The motivation of this study was to test whether machine learning (ML) could be used to process the 3D point cloud data generated by PlantEye sensors to detect individual plants. Individual plant detection is a prerequisite to use complex canopy scans to infer the characters from individual plants. Therefore, this step should expand the portfolio of plant phenotypes that could be inferred from 3D point cloud data. To test the feasibility of such an ML-based pipeline, plant identification was followed by plant counting. Plant counting was prioritized for automation as the current manual counting process is time- and cost-inefficient, and prone to human error. It is, moreover, almost impossible to achieve the automatic counting with current data processing approaches that rely on deterministic prediction models (Kar, Garin et al., 2020; Vadez et al., 2015).

The presented pipeline that automates plant counting consists of five steps. The first three were responsible for trays segmentation, cleaning soil data, filtering out noisy data, and converting the point cloud data to 2D images suitable for the convolutional neural network (CNN) models (Section 2.4 and Section 2.5). The fourth and fifth steps utilized the fine-tuned state-of-the-art object detection models (Faster RCNN Inception-v2 and Faster RCNN ResNet50) to detect and count plants inside the images (Section 4.2). The accuracy of the algorithms developed (based on MAPE–mean absolute percentage error) was 93.18% and 92.87% for the counting of mung bean and chickpea plants, respectively. This could substitute the current manual process, saving substantial operational costs. The design of the pipeline has a high degree of flexibility, which allows any step to be replaced. On top of that, new steps are to be added as research on the utilization of ML-approaches for 3D data is advanced. Importantly, the ML-based algorithms for individual plant detection set a solid foundation for predicting complex phenotypic characteristics based on individual plants.

### 4.2. Use of 3D Point Clouds for Data Preprocessing

A traditional deterministic (non-ML) approach, namely RGS (Region Growing Segmentation), was utilized to preprocess the raw 3D point clouds to separate plants from their surroundings. A key idea behind this algorithm is to merge adjacent points and form point clusters. As soil data were dense, and values very close to each other, they formed the largest cluster (cluster to be filtered). Consequently, the unnecessary background data were eliminated by separating the plant from the soil and eliminating irrelevant object models. The point coordinate information provided by the 3D point cloud data format played an essential role in the segmentation process of raw data. This step improved plant detection accuracy. The CNN model only needed to produce object (plant) proposals within the plant regions instead of the whole image. However, this preprocessing step increased the complexity of the pipeline. Thus, its benefits for data processing need to be carefully considered before implementation. This agrees with the work of, e.g., Muchtar et al. [42], Kim et al. [43], and Mohamed et al. [44], who emphasized the importance of background cleaning in the object detection process. However, despite the fact that 3D point clouds can hold an extra information compared to those that are 2D, there are still several essential shortcomings that currently limit the full exploration of the 3D point cloud data format for purposes similar to ours, i.e., plant detection and counting.

The main challenge in using 3D point clouds for plant detection—or any other feature extraction for that matter—was its sparse, unordered, and unstructured format. Neural network models such as CNN and RCNN are not built to use the point cloud as a direct input. CNNs are a special kind of deep neural network designed to process 2D data (images). The primary purpose of a CNN is to learn the characteristics of the data by recognizing patterns in the images.

The current ML approaches proposed for learning 3D point cloud data have significant limitations. The approaches can be divided into three main categories: view-based, voxel-based, and point cloud-based [42,45,46,47]. One of the early approaches for applying traditional NNs to 3D data is the view-based method. Researchers attempt to project 3D data to 2D images from different angles to be able to use standard 2D CNN for feature extraction [45,47]. Although the view-based method solves the structuring problems of 3D data, it might result in unsatisfactory performance for complex problems. Voxel-based convolutional neural networks (used as 3D-CNN) are another alternative to applying voxelization to solve unstructured and unordered data problems [46]. However, compared to the 3D point cloud, the resolution of the voxel data is low, which can result in a loss in data representation. There are severe shortcomings in voxel-based methods. Researchers have tried to find a method to process raw data. In 2017, a point cloud-based method performed directly on point cloud data was proposed, e.g., PointNet [46]. PointNet uses symmetric functions to solve the ordering problem of point clouds. It uses multilayer perceptron (MLP) to extract features from each point independently and max-pooling layers to extract a global feature to aggregate the information from all points. Therefore, and as the name of the original work suggests (“PointNet: Deep Learning on Point Sets for 3D Classification and Segmentation”), the PointNet algorithm was found to be more appropriate for Semantic Segmentation and Classification tasks, but not for object detection. To sum up, while the techniques have been conceptualized, many of these studies have been performed on a few common CAD (Computer-Aided Design) datasets. Consequently, object detection tasks in a real-world environment remain challenging, i.e., choosing the most appropriate approach for practical applications.

Additionally, in comparison to labeling 2D images, handling and labeling 3D point clouds is complicated. Software dedicated to labeling 3D data is not well established yet. Although current public datasets provide both outdoor and indoor scenes, their usability is limited. As a result, while 2D deep learning-based methods are being successfully utilized for most benchmark-based assessments, there are limited resources of public standard NN models for 3D point cloud data. Improved neural networks for the solution of 3D point cloud problems can be expected to be designed in the coming years [48]. In our study, the 3D point cloud reflections of plant canopies generated by the current optical system (PE(R); F300) might have suffered from partial overlapping. This would be a more frequent case for larger, ‘bushy’ types of plant canopy. Dimensionality reduction, therefore, provided us with the means with which to normalize for the possible differences in 3D-canopy reflections between genotypes with very different structures.

Therefore, to achieve the objectives of this study under the current state of CNN models development, the 3D data were converted to 2D projections, which also substantially simplified the computational requirements by reducing the data size from 3D (width X length X height) to 2D (width X length). It is important to note that height was added to these 2D projections as color (see Section 2.5), which added valuable information and enabled CNN models to better learn plant patterns (the lower large leaves are blue, the upper small leaves are red; see Figure 8). Moreover, the conversion to 2D delivers a “true” projection instead of a skewed projection with conventional 2D imaging devices where the object’s distance to the scanner will skew the actual plant area it captures. In the future, 3D to 2D projections can be generated from different angles and can be fed to standard 2D CNN pipelines. In addition, laser-line scanning technology can be more easily automated in such big systems compared to 2D imaging systems.

Although sensor-based technology is rapidly advancing and can reflect objects with more resolution and details (including the Phenospex-made PE (R) technology; https://phenospex.com/products/plant-phenotyping/planteye-f500-multispectral-3d-laser-scanner/; accessed on 25 August 2021), the advancement of ML-based methodologies to extract features from such 3D information is lagging behind. Even for the investigated dataset, any currently existing 3D algorithms are not likely to provide satisfactory results compared to more advanced 2D algorithms that have proven their effectiveness and robustness on well-known real-world datasets (e.g., COCO—Common Objects in Context). In any case, further experiments will be conducted, and more algorithms tested to extract more of the plant features. Any such advancements could be readily incorporated into the pipeline structure proposed herein.

### 4.3. Implemented CNN Models for Data with Reduced Dimensionality to Recognize Individual Plants and Count Them

Considering the maturity of CNN for the treatment of 3D point clouds, the individual plant recognition and counting process, which was performed as an initial task in our research, was based on the 2D-CNN algorithms. Adapted CNN architectures (Inception and Residual Network (ResNet)) were utilized in this study. The Inception architecture was introduced as GoogLeNet (Inception-v1; [41]) to perform the required CNN operations in parallel. In similar studies, the basic Inception model has been refined in various ways, such as introducing batch normalization [49] and additional factorization ideas [50]. Thus, the Inception model acts as an efficient multilevel feature extractor and has proven its ability to win the ImageNet Large-Scale Image Recognition Competition in 2014. On the other hand, in deep learning algorithms, it can be considered that the performance will increase as the number of layers increases in the model. Nevertheless, in practice, when deeper networks start to converge, a disruption problem occurs. ResNet addresses the degradation problem by introducing a deep residual learning framework [51]. This model consists of basic residual blocks that provide a shortcut connection between layers. This shortcut connection makes it possible to train up to hundreds of layers without worrying about the vanishing gradient problem.

In our case, applying these methods (Inception and ResNet) for detection and counting of mung bean and chickpea plants produced relevant accuracies (6.82% and 7.12% MAPE values, respectively). The mean Average Precision (mAP) scores of the Inception-v2 and ResNet50 models were greater than 0.70 for both datasets. These achieved accuracies are high enough to justify the incorporation of the algorithms in the current mung bean and chickpea processing pipelines to begin saving time and human resources and standardize the operations.

Finally, specific environments and data types require the selection of appropriate algorithms considering the unique characteristics of the problem that will be addressed. No default workflow can be applied to data generally without adjustments according to image detection and processing strategy. Therefore, special efforts are still required to develop the best appropriate application according to the dataset’s characteristics. In this regard, this study provided a simple and efficient pipeline as a baseline for future improvements. As the developed pipeline consists of independent steps, as illustrated in Figure 2, it is easy to modify or extend the pipeline by automatically attaching up-to-date 2D or 3D ML approaches to find different phenotypic features.

## 5. Conclusions

The aim was to detect the individual plants from complex 3D canopy scans generated by PlantEye^®^ sensors as a prerequisite for future exploration of the complex plant features (e.g., canopy architecture). We found that machine learning (ML) would be a suitable approach to detect individual plants. In this work, we deployed this approach for the automation of plant counting tasks, which achieved the accuracies relevant to substitute manual operations that are currently time- and cost-inefficient and prone to human error.

The main contributions and novelty of this study can be summarized as follows. We proposed solving the deficiencies of current ML algorithms for 3D data by using height information together with 2D images and state-of-the-art computer vision models. Specifically, a combination of the 3D point cloud data with state-of-the-art 2D CNN models was proposed. The benefits of the soil segmentation algorithms (RGS and RANSAC) to segment plant data from the background were demonstrated. Finally, a novel and robust pipeline was proposed to process the 3D point cloud data.

Further research will be designed to broaden the use of the ML baseline set in this study. In particular, we will expand the analysis across plant growth stages and investigate the applicability of the algorithms for the detection of additional species. Consequently, we will investigate the options for estimating additional plant features (e.g., branching and individual leaf count and expansion). Furthermore, the potential of improved 3D models using data generated by the newer Phenospex equipment will be explored.

Source code of the proposed pipeline and plant detection, including the following updates, has been published in the following Github repositoriy https://github.com/serkankartal/Machine_Learning_Based_Plant_Detection_on_3D_Canopy_scans; accessed on 9 November 2021.

## Figures and Tables

**Figure 1 sensors-21-08022-f001:**
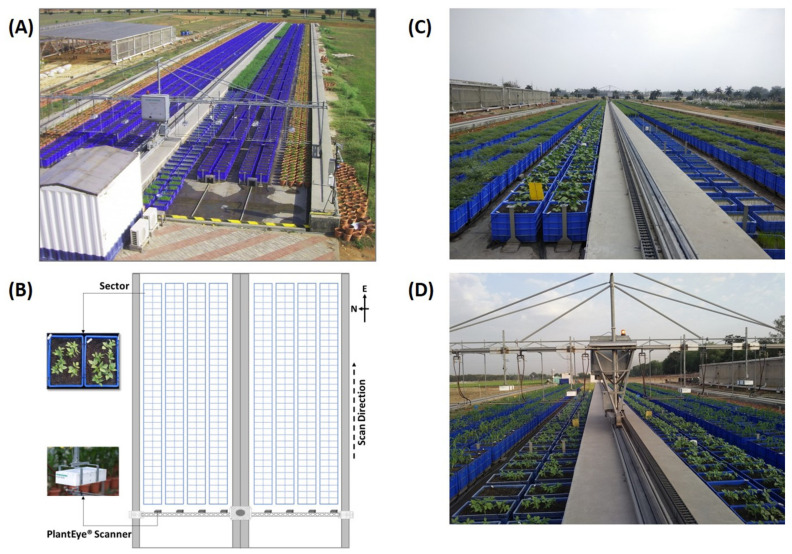
(**A**) Aerial view of LeasyScan high-throughput plant phenotyping facility at ICRISAT; (**B**) Layout of the platform showing sectors and PlantEye^®^ laser scanners that generate a digital reflection of plants in the form of 3D point clouds; and (**C**,**D**) Mung bean and chickpea genotypes monitored continuously for crop canopy traits.

**Figure 2 sensors-21-08022-f002:**
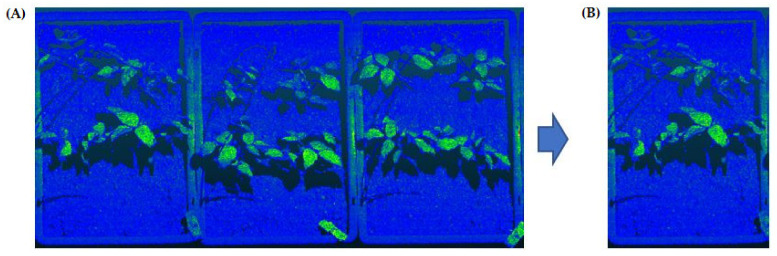
A partial view of a mung bean point cloud data (**A**) and a view of a separated tray sample (**B**). Each data file was divided into 12 (number of trays) equal parts along the *y*-axis to separate the trays from each other.

**Figure 3 sensors-21-08022-f003:**
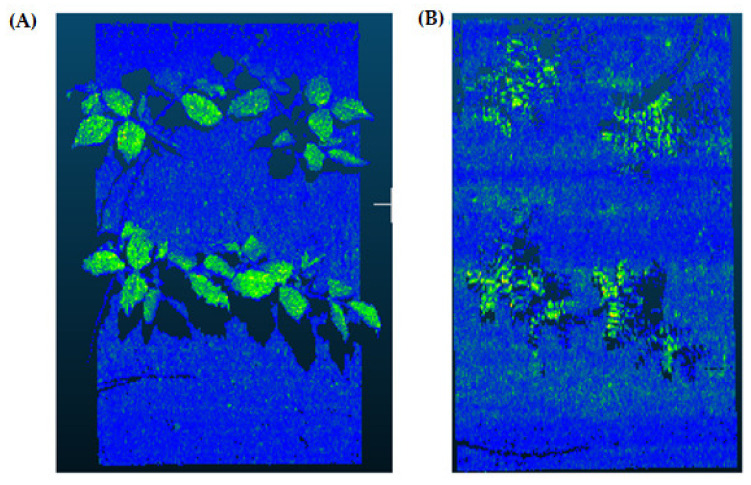
Outputs of tray segmentation process for mung bean (**A**) and chickpea (**B**) files. This operation is the first step of the background cleaning process and is carried out according to the fixed tray dimensions and coordinate information. As a result of this operation, the tray data are completely cleared from the raw data.

**Figure 4 sensors-21-08022-f004:**
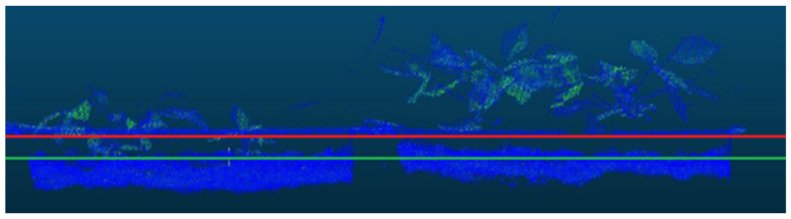
A partial view of a sample raw data with two soil separation lines. The red line points to the selected fixed height coordinate to separate the whole soil data, while the green line points to the other coordinate to keep as much plant data as possible. In the case of using the coordinate value pointed with the red line, it is seen that many plant data belonging to the left tray is lost. On the other hand, if the green line points to the coordinate, many soil data belonging to the correct tray cannot be cleared.

**Figure 5 sensors-21-08022-f005:**
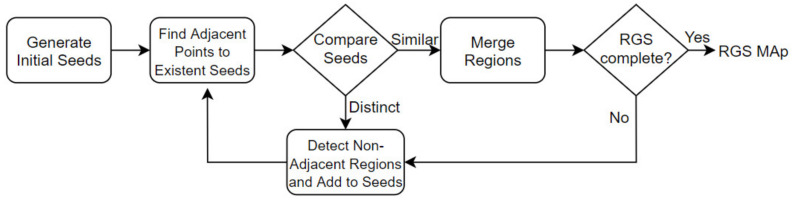
Flowchart of the Region Growing Segmentation. A key idea of this algorithm is to merge adjacent points and form the point clusters. As soil data are dense and very close to each other, they form the largest cluster (the cluster to be filtered) as a result of this process.

**Figure 6 sensors-21-08022-f006:**
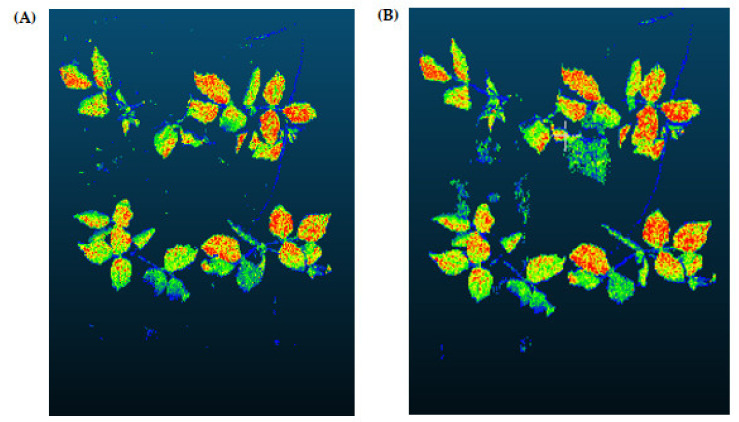
The output of Region Growing Segmentation process for mung bean (**A**). This operation was carried out to increase the accuracy of the proposed pipeline and to minimize false detections. The outputs of a RANSAC Segmentation process for mung bean (**B**).

**Figure 7 sensors-21-08022-f007:**
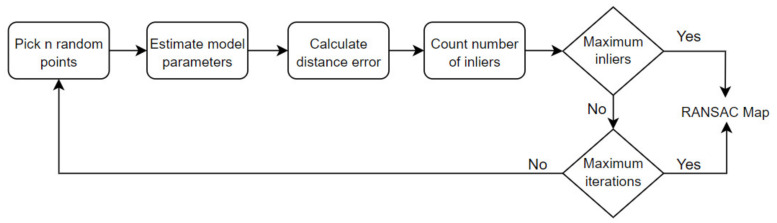
Flow chart of the RANSAC algorithm. A key idea of this approach is to find a mathematical model that covers most of the data. As the amount of soil data is much higher than the plant data, the algorithm finds the model that best fits the soil data. Consequently, the data group that fits the model is found and separated as soil data.

**Figure 8 sensors-21-08022-f008:**
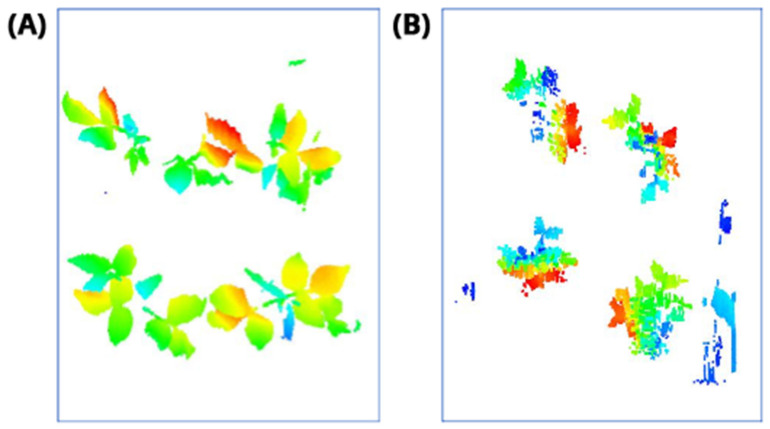
Outputs of Rasterization process—mung bean (**A**), chickpea (**B**). In this process, 3D point cloud data are converted into 2D images according to their height values. Shades of red represent data with a higher height value, medium heights are green, and the bottom ones have shades of blue.

**Figure 9 sensors-21-08022-f009:**
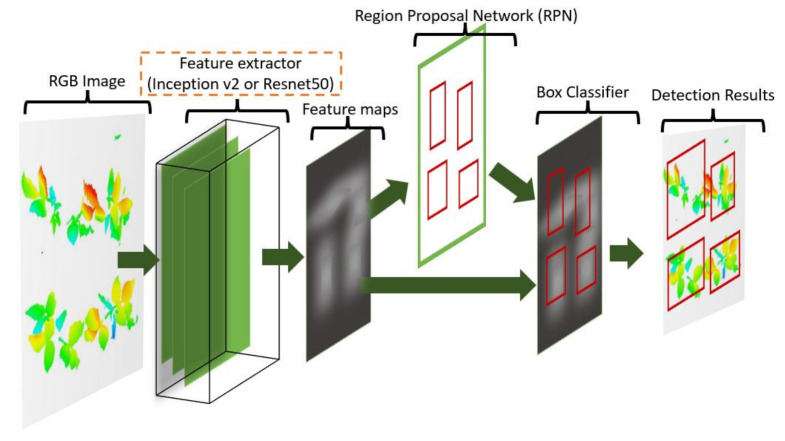
Faster RCNN architecture with Inception-v2 or ResNet50 feature extractor. This architecture uses a region proposal network (RPN) responsible for proposing bounding boxes containing plants. After the training process, the model can detect and label plants on the input image.

**Figure 10 sensors-21-08022-f010:**
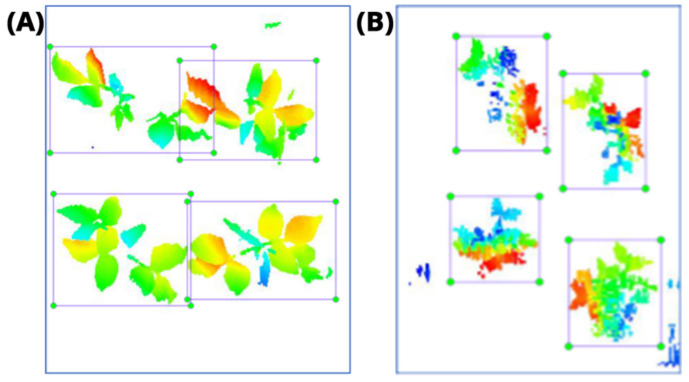
Samples of labeled images – mung bean (**A**), chickpea (**B**). The labeling process is carried out by drawing a box around each plant and labeling the box with the plant name. Experts manually labeled all rasterized images to generate training, validation, and testing datasets.

**Figure 11 sensors-21-08022-f011:**
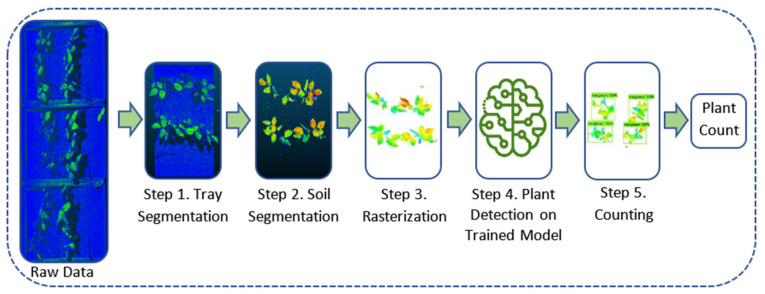
Flowchart of the plant counting pipeline. This pipeline concatenates the 3D data segmentation process and the 2D supervised ML algorithms as one unified model that can be utilized as an end-to-end solution. Accordingly, the developed pipeline can read raw point cloud data files and extract rasterized images of each tray (with the plants detected on it) along with the total number of plants.

**Figure 12 sensors-21-08022-f012:**
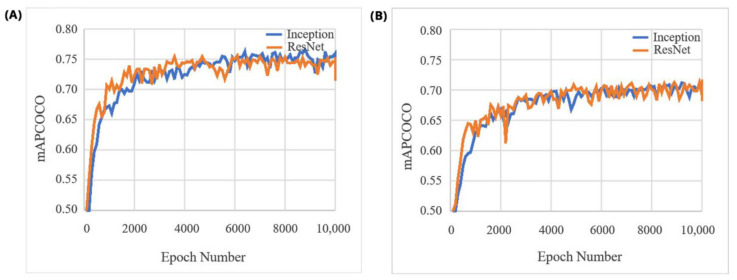
*mAP*_COCO_ scores of the Inception-v2 and the ResNet50 models on mung bean (**A**) and chickpea (**B**) validation dataset during the training process.

**Table 1 sensors-21-08022-t001:** *mAP*_COCO_ values for the COCO dataset and *mAP*_COCO_ for the mung bean and chickpea datasets.

Algorithm	COCO *mAP_COCO_*	Bean *mAP_COCO_*	Chickpea *mAP_COCO_*
Faster RCNN Inception-v2	0.28	0.7525	0.7108
Faster RCNN ResNet50	0.30	0.7477	0.7075

**Table 2 sensors-21-08022-t002:** Ground truth (manually counted) plant counts predicted plant counts and MAPE values for the mung bean and chickpea datasets.

	Test Images (Tray Count)	Plant Count			MAPE	
		Manual Counting	Inception-v2 Prediction	ResNet50 Prediction	Inception-v2	ResNet50
Mung bean	72	275	287	303	**6.82**	11.80
Chickpea	237	1606	1547	**1620**	8.17	**7.13**

**Table 3 sensors-21-08022-t003:** Faster RCNN Inception-v2 and Faster RCNN ResNet50 models produced mung bean and chickpea detection results for randomly selected eight test images and their original views.

	Inception-v2	ResNet50	Ground-Truth
1	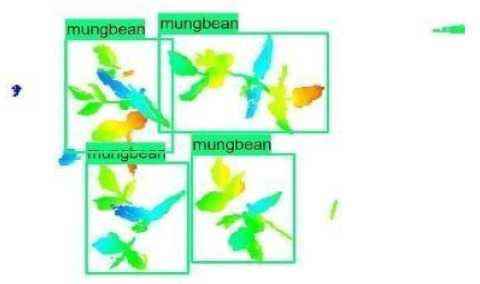	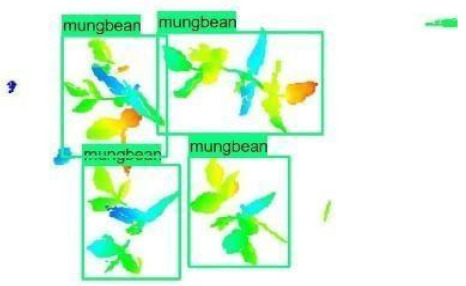	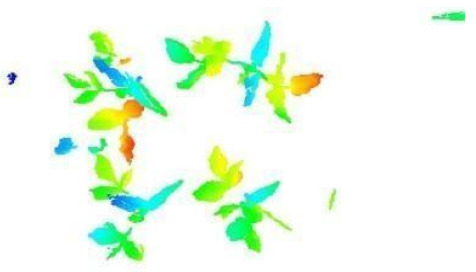
2	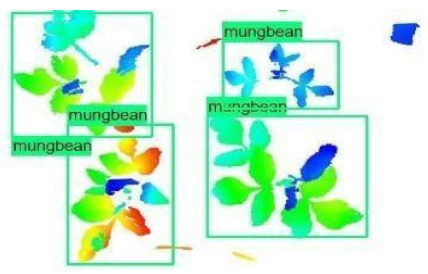	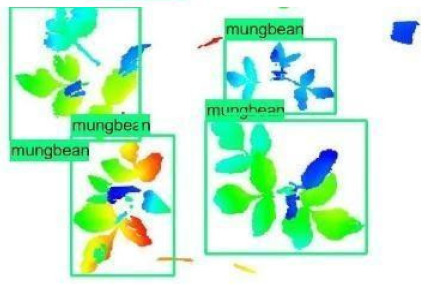	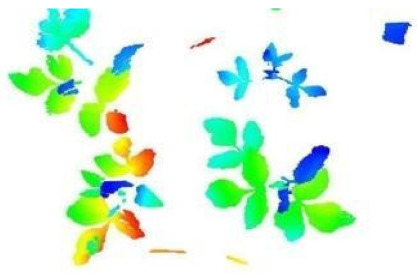
3	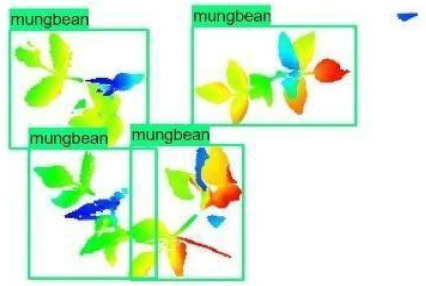	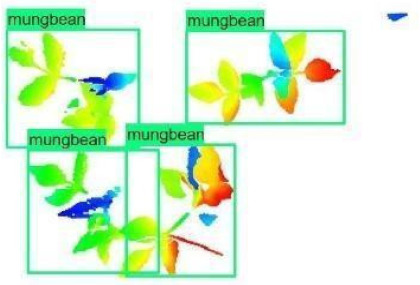	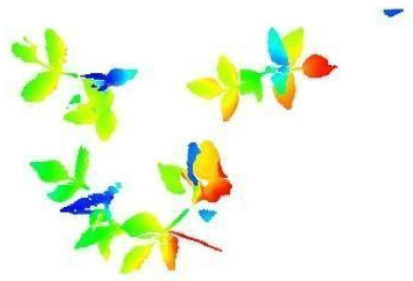
4	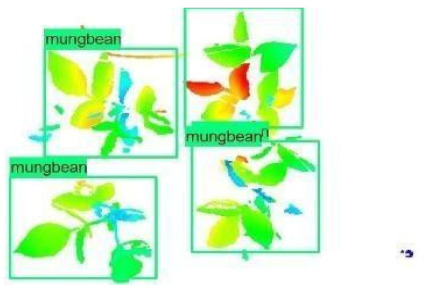	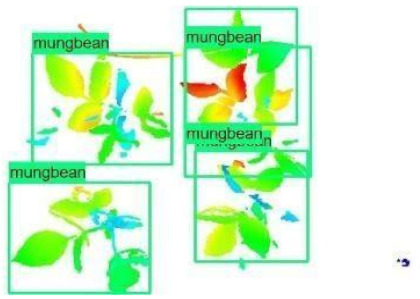	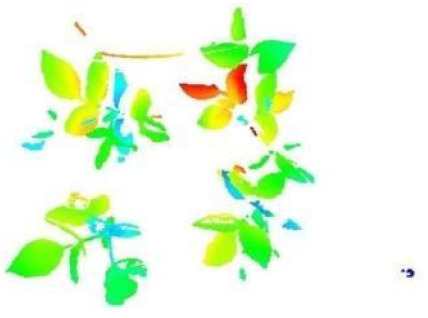
5	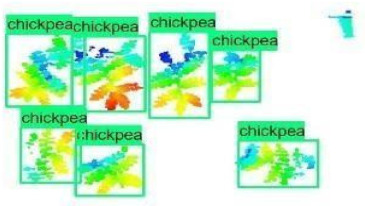	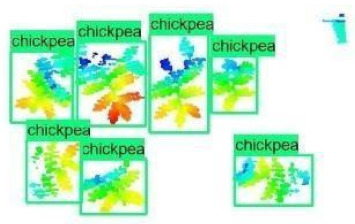	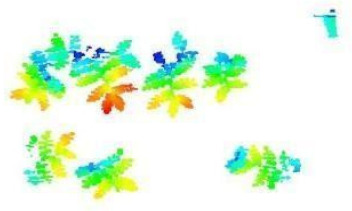
6	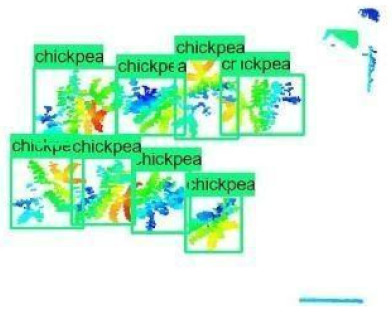	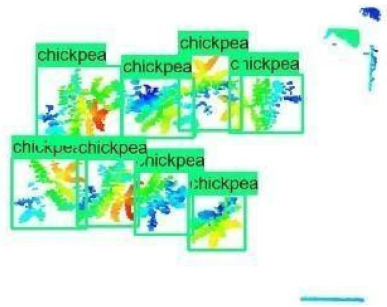	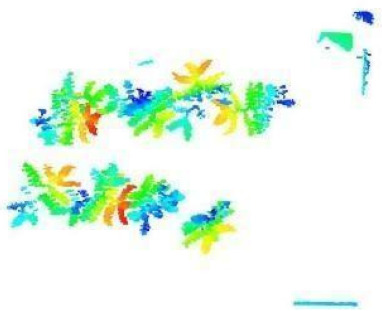
7	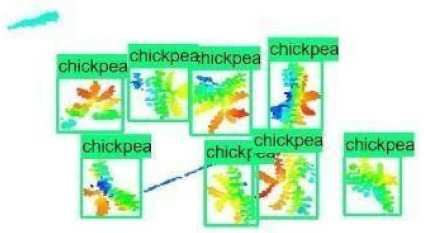	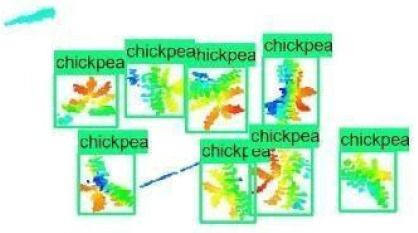	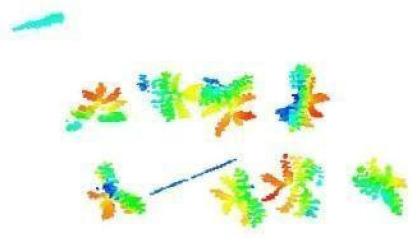
8	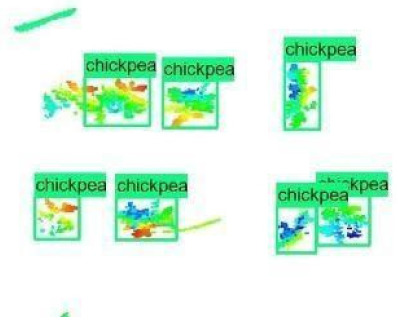	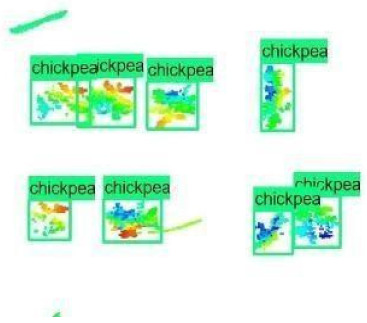	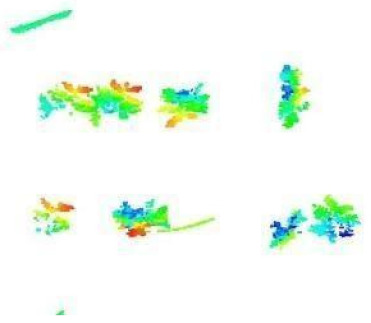

**Table 4 sensors-21-08022-t004:** Training time and plant counting time for the Faster RCNN Inception-v2 and Faster RCNN ResNet50 models. The plant counting time includes all the time required along the proposed pipeline for a raw mung bean data file consisting of 12 trays.

	Training Time	Plant Counting Time
**Faster RCNN Inception-v2**	1.25 h	15 s
**Faster RCNN ResNet50**	1.45 h	18 s

## Data Availability

Source code of the proposed pipeline and plant detection, including the following updates, has been published in the following Github repositoriey https://github.com/serkankartal/Machine_Learning_Based_Plant_Detection_on_3D_Canopy_scans; accessed on 9 November 2021.

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
