# Peer review of "Machine Learning-Based Plant Detection Algorithms to Automate Counting Tasks Using 3D Canopy Scans"

_sensors, 2021, doi:10.3390/s21238022_

Round 1

Reviewer 1 Report

This paper proposes a practical method to segment individual plants from 3D point-clouds. The data and presentation are good, and Discussion is comprehensive enough. However, the following issues should be tackled:

Please Check whether Figure 13 and Figure 14 are technically shown.

The network and method used in this paper are already known. Please clearly demonstrate what aspects of methodology the authors have developed.

In Introduction, the authors say there are only a few studies regarding the extraction of single plant traits from 3D point-clouds, but what are the differences between these previous studies and this research? What limitations have been dealt with based on these studies?

How fast (what efficiency) is this detection method?

It would be better to use passive voice and past tense throughout the paper. Please have multiple checks on English.

Author Response

Dear reviewer, thank you for your feedback. Firstly, we have marked only the most significant changes in the manuscript by revisions. We did extensive English language editing. We also passed it to a native speaker. So, keeping all minor edits in revisions would make the manuscript very difficult to read. Please find answers to your comments below.

Please Check whether Figure 13 and Figure 14 are technically shown.

  • We could see them. If the problem persists, we can communicate that with the editorial office. The images are shown below. Additionally, addressing other comments, we have merged the figures into one. It better illustrates the difference between the models.

The network and method used in this paper are already known. Please clearly demonstrate what aspects of methodology the authors have developed.

  • We emphasized the fine-tuning of the ML models and how the methods were put together in the pipeline. We also stressed the use of height information of the 3D point clouds data. Please, see mainly section 2.6.

In Introduction, the authors say there are only a few studies regarding the extraction of single plant traits from 3D point-clouds, but what are the differences between these previous studies and this research? What limitations have been dealt with based on these studies?

  • We have updated the Introduction section. We referred to more studies and mentioned their limitations and how we are addressing them. Also, the discussion section has been enhanced.

How fast (what efficiency) is this detection method?

  • This is addressed at the end of Section 3. We have added a table summarizing the times needed for training and counting.

It would be better to use passive voice and past tense throughout the paper. Please have multiple checks on English.

  • We agree and have made adjustments throughout the MS to ensure it reads in a more passive manner

Reviewer 2 Report

This study explores the possibility to automatically count plants while extracting more complex features using. The study seems rather premature for publication and is more or less a report which is not necessarily adding to the scientific community. The first part of the introduction is lacking in citations to back up arguments and an overall statement in how this study is contributing something novel.

There is also insufficient information in how the CNN-approach was implemented in terms of the configuration of hidden layers, hyper parameters and so forth. It seems as though the authors have implemented the CNN as a black box without insight into how the NN was optimized. Was it optimized? See "Free Lunch Thereom".

The study is also lacking in terms of figures. Most of the figures look like quick screenshots which do not necessarily contribute to the papers overall clarity and scientific method.

Sorry to be harsh, this paper has potential however requires more work.

Author Response

Dear reviewer, thank you for your feedback. Firstly, we have marked only the most significant changes in the manuscript by revisions. We did extensive English language editing. We also passed it to a native speaker. So, keeping all minor edits in revisions would make the manuscript very difficult to read. Please find answers to your comments below.

The first part of the introduction is lacking in citations to back up arguments and an overall statement in how this study is contributing something novel.

  • We updated the Introduction and added new citations. We also addressed the studies using 3D point clouds and their limitations. Hopefully, the novelty is better highlighted now.

There is also insufficient information in how the CNN-approach was implemented in terms of the configuration of hidden layers, hyper parameters and so forth. It seems as though the authors have implemented the CNN as a black box without insight into how the NN was optimized. Was it optimized? See "Free Lunch Thereom".

  • We clarified the use of the CNN-approach – fine-tuning and parameters set up in Section 2.6, especially 2.6.3.

The study is also lacking in terms of figures. Most of the figures look like quick screenshots which do not necessarily contribute to the papers overall clarity and scientific method.

  • We have edited the Figures. Some were removed. We also edited descriptions, so they are now self-explanatory. For example, Figure 1 was re-illustrated. Figure 3 was removed, and only one specie was kept here. In Figure 7, we merged two images to illustrate the difference between algorithms. Figure 9 has been added newly.

Sorry to be harsh, this paper has potential however requires more work.

  • We did extensive revision and hope it is much better now.

Reviewer 3 Report

Thanks for the opportunity to read this paper.

The work uses artificial intelligence techniques (Machine Learnig) to separate and analyse individual mung bean and chickpea canopy plants from a point cloud captured by LiDAR sensors in laboratory conditions.

It proposes a methodology that from the point cloud captured by LiDAR, the segmentation of the point cloud (separating the data from each pot) is performed, the soil data is removed and two different techniques are evaluated. Once the point clouds corresponding to the plants are isolated, they are rasterised and the plant detection and identification algorithm is applied, evaluating the goodness of predictions.

From my point of view, the methodology used seems to be very appropriate, the experimental design is adequate, the results show the goodness of the methodology and the conclusions endorse the quality of the research.

The methodology applied seems to me to be adequate for the achievement of the research objectives. Maybe, but it would be the subject of another research to apply a neural network for 3D clouds directly as Pointnet

The autors could have added a graph showing the accuracy and precision as a function of the training epochs for their data.  Please include any additional comments on the tables and figures.

Author Response

Dear reviewer, thank you for your feedback. Firstly, we have marked only the most significant changes in the manuscript by revisions. We did extensive English language editing. We also passed it to a native speaker. So, keeping all minor edits in revisions would make the manuscript very difficult to read. Please find answers to your comments below.

The methodology applied seems to me to be adequate for the achievement of the research objectives. Maybe, but it would be the subject of another research to apply a neural network for 3D clouds directly as Pointnet

  • We have emphasized why machine learning for 3D data was not used. We also addressed the studies using 3D point clouds and their limitations.

The autors could have added a graph showing the accuracy and precision as a function of the training epochs for their data.  Please include any additional comments on the tables and figures.

  • It seems the figures were not shown due to technical reasons. Please see Figure 12.

Round 2

Reviewer 2 Report

This reviewer has no more comments.